# Cost of Blood and Body Fluid Occupational Exposure Management in Beijing, China

**DOI:** 10.3390/ijerph17124192

**Published:** 2020-06-12

**Authors:** Daifang Wang, Yan Ye, Qiang Zheng

**Affiliations:** 1Department of Industrial Engineering and Management, Peking University, Beijing 100871, China; wangdaifang@pku.edu.cn; 2Center for Pharmaceutical Information and Engineering Research, Engineering School, Peking University, Beijing 100871, China; 3Department of Occupational Health, Beijing Center for Diseases Prevention and Control, Beijing 100013, China

**Keywords:** cost, blood-borne diseases, occupational exposure, blood and body fluid exposure, exposure management

## Abstract

(1) Objective: The aim of this study was to determine the cost of blood and body fluid (BBF) occupational exposure management in healthcare facilities in Beijing, China. (2) Methods: A survey was conducted from August to October 2018, seeking general information concerning the management of occupational exposure to BBF and the cost of the management process. In total, 216 healthcare facilities were surveyed, using a stratified-selection method. The collected information included BBF management protocols, direct costs such as laboratory testing fees, drug costs and medical service fees, as well as indirect costs, such as wages, lost working time, injury compensation, and psychological counseling time. (3) Results: The cost of post-BBF exposure management varied according to the infection status of the exposure source patients, the immune status of exposed employees, and the location and level of healthcare facilities. The mean values of management cost were determined to be hepatitis B (HBV)-positive source (RMB 5936/USD 897), hepatitis C (HCV)-positive source (RMB 5738/USD 867), *Treponema pallidum* (TP)-positive source (RMB 4508/USD 681), human immunodeficiency virus (HIV)-positive source (RMB 12,709/USD 1920), and unknown sources (RMB 7441/USD 1124). The survey also revealed that some healthcare facilities have insufficient post-exposure management. (4) Conclusions: A better post-exposure management system is needed in Beijing to reduce both infection risk after exposure and costs.

## 1. Introduction

Healthcare workers are at high risk of blood and body fluid (BBF) exposure that facilitates pathogen infection [1]. The major pathogenic microbes that could be transmitted through BBF exposure include the human immunodeficiency virus (HIV), hepatitis B virus (HBV), hepatitis C virus (HCV), and *Treponema pallidum* (TP) [2]. Sharp injury-caused BBF exposure contributes to approximately 40% of HBV and HCV infections in healthcare workers and 2.5% cases of HIV transmission [3]. These work-related injuries not only harm medical workers’ health, but also result in other adverse impacts, such as great costs of post-exposure healthcare management, a shortage of medical staff, and emotional distress to medical staff [4,5,6,7,8]. Therefore, the identification and implementation of administrative approaches that are effective, to prevent and protect healthcare personnel from BBF exposure, are essential to reduce the exposure rate.

Many countries have assessed the costs and benefits of potential protective approaches against BBF exposure. Based on these studies, their governments implemented laws and regulations to enforce preventive methods [9,10]. In China, although guidelines for the prevention of occupational BBF exposure have been proposed (see Appendix A), governmental regulations have yet to be established through legislative procedures to enforce the management of BBF occupational exposure [11,12,13,14,15]. Consequently, the incidence of BBF exposure among healthcare staff is significantly higher in China than in other countries with specific governmental rules [16,17,18,19]. 

The cost/benefit studies in other countries are not directly applicable to China, due to the fundamental differences in procedures, prices and health insurance policies that are related to post-exposure management costs. Therefore, a new cost/benefit assessment is critical for China to develop a suitable plan to control BBF exposure [20,21,22,23,24]. Indeed, very limited research has been performed to determine BBF occupational exposure cost in China. In 2018, Zhang and colleagues estimated that one needlestick injury would trigger a post-exposure management cost of RMB 515 (USD 78) in China, but this result was based on data from foreign countries and eight opinion leaders [19], and thus lacked data of China’s healthcare system. In 2019, Zhao and colleagues reported that one insulin injection-related needlestick injury costs between RMB 1884 (USD 285) and RMB 2389 (USD 361) [25]. However, this study only measured insulin injection-related injuries to nurses and only self-reported direct costs from exposed personnel, which do not accurately reflect the total cost of exposure management. The purpose of the current study was to estimate the cost of BBF post-exposure management in Beijing, China.

Unlike many Western countries, where most healthcare facilities are private healthcare facilities, government-owned public healthcare facilities, including clinics and hospitals, are the major healthcare services in China. China’s healthcare facilities are stratified into three levels, i.e., primary, secondary, and tertiary healthcare facilities [26]. Primary healthcare facilities provide basic preventive, medical and rehabilitative services to local communities. Secondary healthcare facilities deliver more advanced medical and health services to multiple communities, and are responsible for limited teaching and research tasks. Tertiary healthcare facilities provide both primary care and to a larger extent, specialty services. These latter healthcare facilities also perform higher education and research tasks [26].

The healthcare systems in China are unevenly distributed throughout the country. Beijing, the capital city of China, possesses the most complete healthcare network, and therefore systemically represents China’s healthcare institutions at all levels. Beijing consists of 16 administrative districts, of which six are located in urban areas and cover a population of 12,088,000 in 1385 square kilometers. The other 10 suburban districts comprise a population of 9,619,000 in 15,073 square kilometers [27]. There are 267 primary healthcare facilities, 90 secondary healthcare facilities, and 74 tertiary healthcare facilities located in the urban areas. There are 170 primary healthcare facilities, 65 secondary healthcare facilities, and 39 tertiary healthcare facilities located in the suburban areas. This large population and these areas are covered by 705 healthcare facilities, which provide a rich source for cost/benefit analysis. In this study, we investigated the current status of BBF management in medical institutes in Beijing, and provided a comprehensive characterization of the factors that contribute to the costs of post-BBF exposure management. Considering the distinct administrative structures and processes in Beijing, we investigated the procedure of BBF post-exposure management of Beijing healthcare facilities in order to provide a comprehensive and specific account of the post-BBF exposure management’s cost structure and its key factors.

## 2. Materials and Methods

### 2.1. Data Sources

Among the 705 healthcare facilities in Beijing, 216 were selected to survey for post-exposure management and costs. The objects included 148 primary healthcare facilities, 31 secondary healthcare facilities and 37 tertiary healthcare facilities. 

### 2.2. Study Preparation

The survey was jointly designed by specialists from the Beijing Center for Disease Prevention and Control (CDC), representatives from each of the 16 district CDCs in Beijing, healthcare staff from the Department of Nosocomial Infection of Haidian Hospital, and a research group at Peking University. A pre-survey was conducted, via face-to-face interviews, at five healthcare facilities in Beijing to finalize the survey questions, the report form, and the data collection method. 

### 2.3. Selection of Healthcare Facilities

The stratified selection of the healthcare facilities was performed to ensure that healthcare facilities at each level were surveyed. The workloads of district CDCs were also balanced across all 16 districts. A tertiary healthcare facility, with more staff and a more established management system than a secondary or a primary healthcare facility, tends to have a greater workload. In addition, an urban district tends to have more tertiary healthcare facilities than a suburban district. Specifically, we decided to select 11 healthcare facilities from each suburban district, and 15 healthcare facilities from each urban district. A minimum of 10% of healthcare facilities at each level were selected per district. The actual healthcare facilities were randomly chosen by the district CDCs. 

### 2.4. Survey Method

A website with an electronic survey function was provided by Beijing CDC. Each district CDC asked their sampled healthcare facilities to complete the survey online, and to provide their policies and protocols for data analysis. In the survey, each healthcare facility supplied a report of post-BBF exposure management, including the pertinent protocols, direct costs and indirect costs [28]. All of the protocols and cost data were provided by their administrative person-in-charge of the nosocomial department or the entire healthcare facility. Post-exposure management included the management of testing, reporting, medication, counseling, and compensation prior to the clearance or confirmation of infection. The treatment of any resultant infection was excluded from the post-exposure management cost [28].

### 2.5. Exclusion Criteria

Healthcare facilities that returned blank, or substantially incomplete surveys were excluded. Healthcare facilities that did not report the protocol or the cost of exposure management were also omitted.

### 2.6. Data Extraction

Post-exposure management protocols from individual healthcare facilities were compiled to build a workflow chart and a cost structure of the post-exposure management process in Beijing. The definitions of post-exposure management costs, including direct and indirect costs, were based on the Guideline for Prevention and Control of Occupational Exposure to Blood-Borne Pathogens (GBZ/T 213-2008) (GB stands for the national standard established by the Chinese government) [28]. Direct costs included the expenses of laboratory testing, medications, and materials (see Appendix A). Indirect costs comprised psychological counseling costs, wage loss, transportation and communication costs, and compensation. The mean cost of multiple healthcare facilities was calculated as the weighted means of the costs weighted by the numbers of the healthcare facilities’ healthcare staff who performed invasive procedures. Lost wages were calculated by multiplying the lost days by the daily wages, as reported in the survey. Post-exposure management cost was calculated in Chinese RMB and then converted to U.S. dollars with a currency exchange rate of 6.62:1, which was the standard of 2018. An inflation-adjusted discount rate of 3% [29] was also used to adjust the costs in comparison with costs from previous years.

### 2.7. Statistical Analysis

The Mann–Whitney U test was used for the comparison between the two independent groups. The Kruskal–Wallis test was utilized for comparison between more than two independent groups. All the analyses were performed using IBM SPSS Statistics 24.0 (SPSS, Chicago, IL, USA) and R.

## 3. Results

This study surveyed 216 healthcare facilities. By the conclusion of the survey in October 2018, 211 valid reports from 211 healthcare facilities were collected, corresponding to a 98% response rate, which represented 30% of all healthcare facilities in Beijing [27]. The successfully surveyed healthcare facilities included 33% (37/113) tertiary healthcare facilities, 17% (26/155) secondary healthcare facilities, and 34% (148/437) primary healthcare facilities in Beijing [30]. The urban districts hosted 62 surveyed healthcare facilities, while the other 149 healthcare facilities were in suburban areas. Table 1 provides detailed numbers for these healthcare facilities. In this survey, post-exposure management costs cover the expenses to physicians, nurses, interns, advanced trainees and cleaning personnel. 

As shown in Table 1, all the tertiary healthcare facilities, irrespective of their location, have established written protocols for post-BBF exposure management, records for past BBF exposure incidence, and a dedicated department of nosocomial infection. In contrast, secondary and primary healthcare facilities exhibited clear dissimilarities that were associated with their urban vs. suburban locations. While 94% (58/62) of urban healthcare facilities established post-exposure management protocols, this protocol was not available in 37% (55/149) of the suburban healthcare facilities. Indeed, a Mann–Whitney U test revealed a significant (*p* < 0.05) difference between these two groups (*p* = 7.51 × 10^−6^, U = 3212, df = 1). Then, the data from the urban and suburban areas were combined to determine the association between the healthcare facility level and the protocol availability. Among the primary, secondary and tertiary healthcare facilities, 53% (79/148), 92% (24/26), and 100% (37/37) of the healthcare facilities had protocols, respectively. A Kruskal–Wallis test showed that the difference among these three levels of healthcare facilities were statistically significant (*p* = 9.60 × 10^−7^, χ^2^ = 27.71, df = 2). We further performed a pair-wise comparison among the three sets of data using a Mann–Whitney U test with multiple comparison correction by the Bonferroni’s method. This analysis identified a significant difference among the tertiary vs. primary healthcare facilities (adjusted *p* = 2.16 × 10^−5^) and the tertiary vs. secondary healthcare facilities (adjusted *p* = 8.50 × 10^−3^), but no significant difference between the primary and secondary healthcare facilities (adjusted *p* = 2.74 × 10^−1^).

Figure 1 summarizes the general post-exposure management process of 151 healthcare facilities with written management protocols. The major components of this process were immediate management, post-exposure prophylaxis (PEP), and follow-up exam(s). Immediate management includes wound treatment and laboratory testing procedures that are performed within 24 h post-exposure. The identification of pathogens that could transmit through BBF exposure determines the specific treatment approach. The laboratory testing measures the antigen/antibody levels in the exposed staff and guides the administration of preventive medicines. A follow-up exam refers to the laboratory testing at a later time point to assess the effect of immediate management and post- exposure prophylaxis.

The capability for laboratory tests, PEP medication plan and follow-up exams varied markedly among the surveyed medical healthcare facilities. As is summarized in Table 2, while a small fraction of healthcare facilities could cover the entire management process, most healthcare facilities partially or completely relied on other healthcare facilities to manage the incidences of BBF exposure. Urban healthcare facilities were more likely to be able to handle the entire management process than were suburban healthcare facilities. The pathogen type was a major determinant factor of management capacity. For instance, while 65 healthcare facilities could process HBV-related exposure, only four healthcare facilities in Beijing could entirely handle HIV-related cases. As most healthcare facilities could only investigate the infection status of inpatients, the pathogen type of outpatients was treated as an “unknown” group in Table 2. 

Based on the above data, we calculated the direct costs to manage post-management exposure for each pathogen. The costs were calculated as the means weighted by the number of healthcare staff who performed invasive procedures in each healthcare facility. When one healthcare facility relies on another to handle the management, all of the related costs will be assigned to the healthcare facility where the injuries originally occurred. The direct costs, as shown in Table 3, cover major expenses directly related to the management process, such as the laboratory testing, preventive treatment, and copayment. The management for HIV-positive exposure is the most expensive, due to the high PEP and follow-up costs. The PEP cost is mainly composed of expensive medications that use reverse transcriptase inhibitors. The HIV follow-up exams are costly because these tests are frequently conducted. The management cost for unknown source exposure is high because all four pathogens are tested in the immediate management step. The range of costs in these healthcare facilities is broad, mainly ascribed to the variation in post-exposure management protocols, testing costs, and medication costs.

The indirect costs of BBF exposure management include counseling costs, lost wages, transportation expenses, communication expenses, compensation, and other miscellaneous costs. Counseling includes both medical and psychological consultation. An accurate measure of the indirect costs is the time spent in exposure management, as shown in Figure 2. The hours of laboratory testing and medication constitute the largest portion of time. HIV exposure management requires more counseling time than other types of exposure do.

Figure 3 calculates the total costs as the sums of the direct and indirect costs. The total costs of HBV, HCV, TP, HIV, and unknown source post-exposure management, were RMB 5936 (USD 897), RMB 5738 (USD 867), RMB 4508 (USD 681), RMB 12,709 (USD 1920) and RMB 7441 (USD 1124), respectively. The total costs for these pathogens were significantly different (*p* < 0.001, Kruskal–Wallis test).

Table 4 shows the post-exposure management costs means calculated from the urban healthcare facilities and the suburban facilities. There is a clear trend for urban facilities to charge more than suburban facilities. The differences in the indirect costs between the urban and suburban facilities are large.

Table 5 shows the post-exposure management costs means calculated from the primary, secondary and tertiary healthcare facilities. The direct costs show slight differences between the different levels of facilities. Indirect costs show large differences between each of the three level facilities. There is a clear trend that tertiary facilities have the largest indirect costs and the secondary facilities have the least indirect costs for all five exposure sources. 

Table 6 shows the entities that are responsible for the cost of BBF exposure management. The costs-sharing distribution for four types of healthcare staff, i.e., healthcare facility employees, medical interns, trainees and cleaning staff, were calculated separately. The employment status of the exposed staff determines the payment structure of their post-exposure management. For all the types of staff, the healthcare facility, the government and the insurance covered at least 60% of the total costs. Significant variabilities existed in how each healthcare facility treats the cost of post-exposure management. However, since both the healthcare facility care and medical insurance are primarily funded by the government, these portions are all directly or indirectly supported by the government. It is worth noting that, although personal co-payments are necessary for treatment, this may discourage some staff from reporting BBF injuries. 

## 4. Discussion

In 2011, approximately 121.3 percutaneous injuries per 100 occupied beds occurred in China. This incidence is far higher than the 19.5 percutaneous injuries per 100 occupied beds per year in the United States. The lack of protective approaches for healthcare staff to reduce BBF exposures has threatened Chinese medical staff’s health and imposed a great economic burden on China [16,17,31]. The establishment of governmental regulations to solve this problem requires a precise assessment of the costs related to post-exposure management, which constitutes a major purpose of this study. 

Here, we investigated the cost structure and its key factors of post-BBF exposure management in Beijing, China. The final costs of post-exposure management that we calculated are similar to the results of a prior study in the United States. In their analysis, the management of HCV and HIV exposure costs USD 650 and USD 2456, respectively [32]. The corresponding costs in our study are USD 867 and USD 1920. 

We elucidated a drastic cost variability among the healthcare facilities of distinct locations and rank levels. The difference between urban and suburban facilities may be because suburban facilities often partially or fully lack management protocol for BBF exposure (see Table 1). 

The variability among facilities of difference ranks is due to indirect cost difference, as is shown in Table 5. The main portion of indirect costs is the wages of the related persons determined by the time expenditure including the management time and paid leave (see Figure 2 and Figure 3). Tertiary facilities exhibit the highest indirect costs for two reasons. First, all of the tertiary facilities have full post-exposure management protocols and they are more likely to undertake the whole process of post-exposure management (Shown in Table 1). Second, most tertiary facilities in Beijing are general hospitals which have the heaviest work load, so the exposed staff need to wait for a long time with other patients for laboratory testing and medication. The primary facilities also have higher indirect costs than secondary facilities. One reason is that most of these facilities do not have the capacity to directly handle the exposures; they spend extra time, such as traveling time, managing their patients to be seen at external facilities. Another reason is that primary facilities are less busy than other facilities and tend to give more paid leave to exposed staff. This reality exists because there are no consistent regulations of paid leave duration among facilities in Beijing. The secondary facilities tend to have the least management time expenditure. The major reason is that secondary facilities in Beijing are mostly specialized facilities that selectively accept certain types of patients. Therefore, the sources of exposure in secondary facilities are simpler and more predictable than in primary and tertiary facilities, which greatly facilitates the exposure management. In addition, these facilities are less busy than tertiary facilities. For secondary facilities that do not accept patients with infectious diseases, they established designated systems to treat their exposed staff, which also reduces management time and therefore the indirect cost. Based on these findings, indirect costs can be reduced by enhancing the exposure management capability for primary hospitals. For tertiary facilities, more efficient exposure management, like quick access medication for exposed staff, can reduce the management time and therefore the costs.

Along with these findings, we identified several weaknesses in current post-BBF exposure management in Beijing, China. 

First, a small fraction of healthcare facilities lacks a post-exposure management protocol and takes no measures to manage these exposures. For healthcare facilities with existing protocols, a large portion do not possess the capacity to fully handle the exposures. Specifically, 43%, 62%, 45%, 69%, and 76% of these facilities are not capable of managing HBV, HCV, TP, HIV, and unknown sources, respectively (see Table 2). 

Second, the referral system for post-exposure management is not yet mature. Some healthcare facilities that do not have the capacity to manage the exposures need to refer their patients to other more specialized healthcare facilities. However, we frequently observed that these healthcare facilities were not aware of where to refer the patients. In particular, only 40% of the surveyed healthcare facilities referred HIV cases to the four healthcare facilities that were designated to handle HIV infection in Beijing. The other 60% either referred the exposed staff to an inappropriate location or made no referral at all. As a result, the HIV-exposed person might not receive timely testing and/or preventive treatment. 

Third, the current system lacks the ability to identify the pathogen source of all patients. Most surveyed healthcare facilities only reported the infection status of inpatients and day surgery patients, but not outpatients, due to the paucity of medical records. Consequently, the immediate management of exposure caused by unknown outpatients is the most expensive, with follow-up ranked as the second most expensive, as shown in Table 3. In addition, some outpatients were not willing to test for disease infection, which prohibits immediate preventive treatment. 

Among these problems, one can see an obvious and urgent need to improve the system to manage BBF-related exposures for healthcare staff. Currently, HIV-related management is especially weak and requires extra attention by the health department of the Beijing government. Here, we propose two approaches to solve the HIV-related issues. Firstly, Beijing needs to equip more healthcare facilities with the capacity to entirely handle HIV-related exposures. At least one such healthcare facility should be present in each of the 16 districts. Secondly, a governmental regulation should be established to ensure that each medical healthcare facility is aware of its designated healthcare facility where HIV-related patients should be referred. 

## 5. Conclusions

In this study, we systematically surveyed post-BBF exposure management costs in Beijing, China. We also identified the key variable factors that contribute to these costs for the first time, such as location and hospital level. We concluded that the current management of BBF occupational exposure is both weak and expensive, and could be significantly improved by governmental regulations to decrease the related economic and social burdens. More effective interventions are called for to prevent and reduce the occurrence of BBF exposure. Additional efforts are also needed to enforce all the healthcare facilities to properly establish and execute post-exposure management protocols, especially in the handling of HIV-related exposures. More convenient and efficient exposure management systems are needed to both benefit the exposed staff and also reduce the costs.

## Figures and Tables

**Figure 1 ijerph-17-04192-f001:**
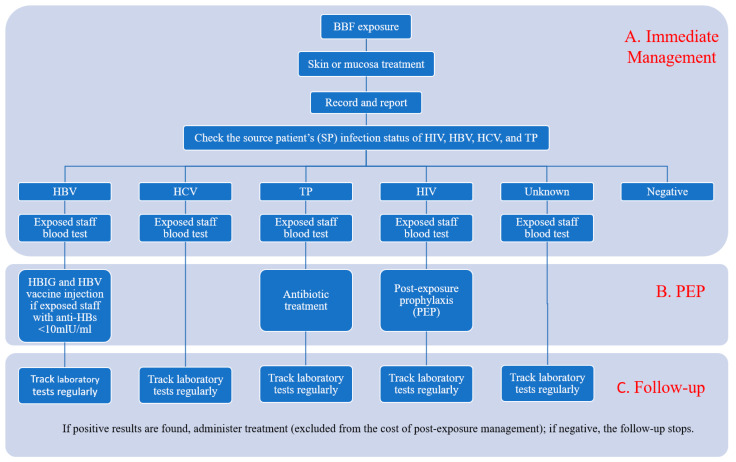
Process of management of occupational exposure to BBF in the healthcare facilities of Beijing, China. BBF: blood and body fluid; HIV: human immunodeficiency virus; HBV: hepatitis B virus; HCV: hepatitis C virus; TP: *Treponema pallidum*; PEP: post-exposure prophylaxis; HBIG: human hepatitis B immunoglobulin.

**Figure 2 ijerph-17-04192-f002:**
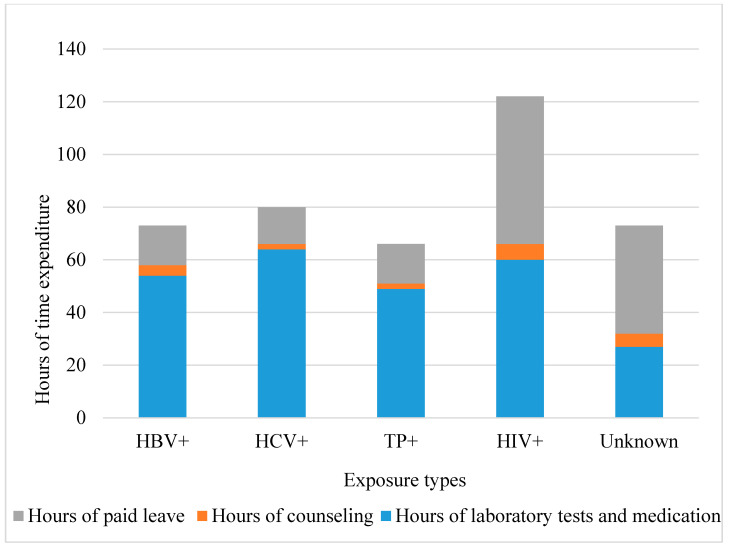
Time expenditure associated with occupational BBF exposure management. HIV: human immunodeficiency virus; HBV: hepatitis B virus; HCV: hepatitis C virus; TP: *Treponema pallidum*. The mean time was weighted by the number of staff with invasive duties.

**Figure 3 ijerph-17-04192-f003:**
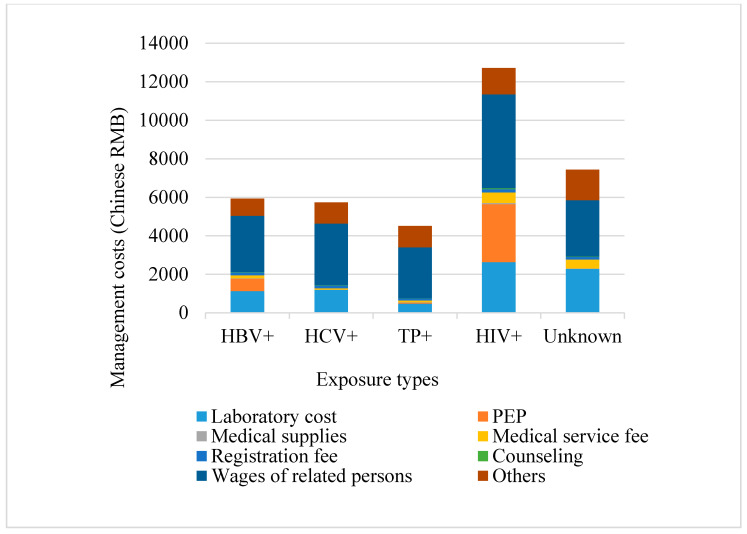
Direct and indirect costs categorized by the different types of medication. HIV: human immunodeficiency virus; HBV: hepatitis B virus; HCV: hepatitis C virus; TP: *Treponema pallidum;* PEP: post-exposure prophylaxis. Mean costs were weighted by the number of staff with invasive duties at each healthcare facility.

**Table 1 ijerph-17-04192-t001:** Characteristics of the surveyed healthcare facilities in Beijing.

	Tertiary Healthcare Facilities		Secondary Healthcare Facilities		Primary Healthcare Facilities	
Location of healthcare facilities	Urban	Suburban	Urban	Suburban	Urban	Suburban
Number of healthcare facilities	27	10	11	15	24	124
Average bed number	697	464	319	250	43	29
Average number of employees	1540	637	576	401	124	69
Average number of beds	319	464	250	250	43	29
% with a protocol for BBF exposure management	100%	100%	100%	86%	83%	57%
% with records of BBF exposure	100%	100%	100%	67%	58%	52%
% with a department of nosocomial infection	100%	100%	100%	87%	75%	60%

**Note:** BBF: blood and body fluid.

**Table 2 ijerph-17-04192-t002:** Capability of the healthcare facilities in Beijing, China, to handle post-BBF exposure management.

Source Patient Infection Status	Execute Entire Management Process		Execute Part of Process		Refer to Other Healthcare Facilities Entirely		No Protocols	
	Urban	Suburban	Urban	Suburban	Urban	Suburban	Urban	Suburban
HBV	27	38	18	37	14	70	3	4
HCV	23	15	15	28	22	100	2	6
TP	12	8	32	63	16	71	2	7
HIV	4	0	24	38	31	103	3	8
Unknown	19	11	8	13	25	101	10	24

**Note:** BBF: blood and body fluid; HBV: hepatitis B virus; HCV: hepatitis C virus; TP: *Treponema pallidum.*

**Table 3 ijerph-17-04192-t003:** Direct costs of the BBF exposure management in Beijing, China (in Chinese RMB).

Source Patient Infection Status	Immediate Management	PEP	Follow-Ups	Total Direct Costs
	Mean ^1^ (95% CI)	Mean (95% CI)	Mean (95% CI)	Mean (95% CI)
HBV	447 (246–693)	648 (451–844)	1013 (563–1462)	2107 (1343–2871)
HCV	302 (182–421)		1116 (679–1554)	1418 (877–1960)
TP	238 (118–358)	59 (33–84)	448 (272–625)	745 (443–1047)
HIV	277 (146–407)	3748 (2378–5118)	2400 (1463–3337)	6425 (4261–8589)
Unknown	683 (390–977)		2218 (1356–3079)	2901 (1778–4024)

**Note:** BBF: blood and body fluid; PEP: post-exposure prophylaxis; HIV: human immunodeficiency virus; HBV: hepatitis B virus; HCV: hepatitis C virus; TP: *Treponema pallidum*; CI: confidence interval; RMB: Chinese yuan. ^1^ Means were calculated from the data of each healthcare facility weighted by the number of healthcare staff with invasive duties.

**Table 4 ijerph-17-04192-t004:** Costs of the BBF exposure management categorized by the locations in Beijing (in Chinese RMB).

Source Patient Infection Status	Direct Costs Mean ^1^		Indirect Costs Mean		Total Costs Mean	
	Urban	Suburban	Urban	Suburban	Urban	Suburban
HBV	2348	1529	4921	2287	7269	3816
HCV	1551	1066	5499	2684	7050	3750
TP	652	780	4966	3443	5618	4223
HIV	6629	5905	6611	4780	13240	10,685
Unknown	3063	2462	5464	3273	8527	5735

**Note:** BBF; blood and body fluid; HIV: human immunodeficiency virus; HBV: hepatitis B virus; HCV: hepatitis C virus; TP: *Treponema pallidum*; RMB: Chinese yuan. ^1^ Means were calculated from the data of each healthcare facility weighted by the number of healthcare staff with invasive duties.

**Table 5 ijerph-17-04192-t005:** Costs of the BBF exposure management categorized by the facility levels in Beijing (in Chinese RMB).

Source Patient Infection Status	Direct Costs Mean ^1^			Indirect Costs Mean			Total Costs Mean		
	Primary	Secondary	Tertiary	Primary	Secondary	Tertiary	Primary	Secondary	Tertiary
HBV	1765	1884	2253	3043	1080	4702	4808	2964	6955
HCV	1187	1250	1515	3199	1075	5275	4386	2325	6790
TP	714	620	785	3237	1188	4882	3951	1808	5667
HIV	6021	6778	6414	4470	3862	6940	10,491	10,640	13,354
Unknown	3093	2957	2840	3504	2087	4671	6597	5044	7511

**Note:** BBF: blood and body fluid; HIV: human immunodeficiency virus; HBV: hepatitis B virus; HCV: hepatitis C virus; TP: *Treponema pallidum*; RMB: Chinese yuan. ^1^ Means were calculated from the data of each healthcare facility weighted by its number of healthcare staff with invasive duties.

**Table 6 ijerph-17-04192-t006:** Percentage of the cost-sharing of the post-BBF exposure management.

Employment Status of Exposed Persons	Facility Care	Government Care	Medical Insurance	Personal Co-payment	Host Facility	Staffing Agency
Healthcare facility employees	60%	5%	19%	16%	0	0
Medical interns	45%	3%	13%	31%	8%	0
Advanced trainees	48%	4%	12%	27%	8%	0
Cleaning staff	46%	2%	12%	17%	0	23%

**Note:** BBF: blood and body fluid. Numbers are the mean percentages of the costs shared by different entities. Means were calculated from the data provided by each healthcare facility weighted by their number of staff with invasive duties.

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
