# Peer review of "Cost of Blood and Body Fluid Occupational Exposure Management in Beijing, China"

_ijerph, 2020, doi:10.3390/ijerph17124192_

Round 1
Reviewer 1 Report
Thank you for the revision.
Author Response
Dear Reviewer,
Thank you very much for your helpful and insightful comments.
The Authors
Reviewer 2 Report
No further comments
Author Response

(The authors gave the same response as above.)

Reviewer 3 Report
I would like to thank the authors for their answers and detailed clarifications to my questions in the first review round. However, I still miss in the results the cost (both direct and indirect) distribution by hospital level (1.ary, 2.ary and 3.ary) and by location (urban vs suburban). I will kindly siuuggest to update the table 3 for this purpose.
Are these cost smilar in the different strats assessed? I will expect some difference. Please consider this and discuss how this will impact the palliative intervention.
Author Response
Please see the attachment.

This manuscript is a resubmission of an earlier submission. The following is a list of the peer review reports and author responses from that submission.
Round 1
Reviewer 1 Report
Comments to authors
In the present manuscript, the authors estimated the costs related to occupational diseases related to exposure to body fluid containing pathogenic agents in Beijing. Overall, the paper reads well however, I have number of issues to address to the authors.
- I miss how the strata investigated were selected. The authors mention that “the method of selection was decided on the basis of both scientific and administrative considerations”. Can you please provide a summary as a table for the scientific and administrative criteria used?
- Can the authors include in the method section the model used for the estimated costs? The parameters involved in the calculation?
- In the Table 1, the numbers of primary healthcare facilities sum up to 148. In the introduction and elsewhere the authors report that they investigated 151 primary healthcare facilities. Can they explain this discrepancy?
- I wonder if workers at risk of exposure to HBV do not get the vaccine before potential exposure occurs.
Reviewer 2 Report
The introduction is well-done, preparing and raising interest in the reader’s mind. The methodology, and statistical analysis is thought-through, and include the important parameters and characteristics. My main objection is the lack of scientific and intellectual novelty in this manuscript. A simple change in the Insurance policy, for instance, can change the numbers and therefore the conclusions.
Reviewer 3 Report
The manuscript considers the costs of blood and body fluid exposure for healthcare staff in different Settings and has a potential to provide important information about the costs of the exposure management. Please could you find below suggestions for further improving the manuscript.
Cost of Blood and Body Fluid Occupational Exposure Management in Beijing, China
Abstract
The objective is clearly described, but it is unclear for the reader why authors consider this should be examined – later it is mentioned that costs are higher than previously reported. This also makes the conclusion(s) difficult to follow – why are the costs considered economic burden? Why should effective post-exposure to BBF be economic burden? Similarly, based on the information provided in the abstract, it is difficult to follow how the authors conclude that some facilities have insufficient post-exposure management. Also – when noted that costs are higher than previously reported – when previously – was inflation considered? While it can be conceivably be expected that the authors mean BBF exposure from patients to healthcare professionals, it would be good to clarify this.
Specific comments abstract:
Line 23: please include the costs in the brackets behinds the specific diseases e.g. TP positive source (¥5,936/$897).
Introduction
The information provided in the introduction is somewhat confusing. The reader is left unsure whether the scope of the manuscript is in prevention of the BBF exposure injuries (and the cost effectiveness) by using the safety devices or in estimating the costs of the post-exposure management and comparing this with previously reported costs.
At the end of the introduction clear aims should be provided – rather than what has been calculated – i.e. what the study aims to achieve.
Line 36: … recommended to be screened in all blood products by… This is somewhat confusing, as this leads to expectation that the manuscript considers infections transferred from blood products. Please consider rephrasing.
Line 51: this information is reported already in line 41.
Line 84: participants – not subjects
Line 81 to 91: Should be largely in methods section
Materials and Methods
Please provide information how the authors decided which infections were included in the detailed costs analyses (e.g. the WHO reference at the start of the introduction?). Also, how the authors decided which are direct and indirect cost following the exposure. How are these defined?
In addition, considering different types of the institutions included in the study, it would be beneficial to provide information (for example in broad categories) of the size of the institutions (number of healthcare professionals working + cleaning staff), types, and whether urban/rural. This information would help in understanding the analyses.
Line 94: Please write the number starting the sentence in letters
Line 97: Please start the sentence with a capital letter – Please consider using specialists in preventive medicine instead of preventive medical specialists.
Line 107: Please could you clarify why different districts were asked to select randomly 15% - 35% of the establishments - not, for example, a fixed number of facilities.
Line 114: Please add space above
Line 118: Was information collected regarding different occupational groups?
Lines 126 & 128: Please consider transferring to statistical methods
Line 131: weighted means for what?
Line 132: Please be considerably more specific – which groups were compared? Please specify which statistical tests were performed (i.e. which group comparisons). What p-value was used, was p-value adjusted for multiple comparisons. Was qualitative methods used e.g. in evaluation of protocols? What about descriptive statistical methods?
Results
Please correct the reporting of the significance tests (e.g. K-W) to include χ2, p-value, and df
Line 137: Please correct the percentages e.g. 211 (98%) of the surveyed 216 healthcare facilities…
Line 138: Please write number in letters when at the beginning of the sentence. Also, the information should not be completely duplicated both in the text and in the table. As currently presented the information is difficult follow in the text. Please consider redrafting.
Line 140: “22” – what does this number refer to? Were the information collected only from the 22 institution, if so, this was not clear from the methods section?
Line 144: location differences – please consider using "differences between urban and rural…"
Line 146: The results are difficult follow. Please clarify, does that mean that when all facilities were included there were significant differences between rural and urban? Urban facilities reported more frequently having a protocol in place? Same with second set of the results.
Line 149: please consider rephrasing – as the text is currently written, it appears that authors’ judge the quality of the protocols rather than the frequency of which the protocols are available.
Figure 1: is this based on the standard practice or compiled from the protocols included in the study? While the figure provides excellent overview of the process, the methodology (analysis) used to its development should have been included in the methods section.
Table 2: Please provide the figures divided between rural and urban
Discussion and conclusions
There is certain disconnect between the results and large parts of the discussion. While the results concentrated on estimating the costs of single episodes of exposure and its management, the discussion draws in more system wide elements such as reducing the number of exposures, for example by using the safety devices. However, as the manuscript does not provide information about these aspects, the reader cannot readily follow the discussion and it becomes removed from the stated purpose i.e. estimating the costs.
Line 235: dot is missing at the end of the sentence.
Line 237: Here the discussion becomes removed from the results presented in this manuscript. As the purpose of the manuscript was to estimate the costs of the post exposure treatment, not the burden of the treatment, the maniscript should concentrate in discussion and interpretation of the results from the current analyses. (e.g. manuscript did not provide current evidence regarding e.g. number of sharps injuries within the included facilities).
Line 242: This information was also not comprehensively included in the results section, hence the reader has a difficulty to follow the argumentation.
Round 2
Reviewer 3 Report
Thank you for having the change to re-read the manuscript. Authors have addressed the issues raised previously with considerable care and diligence. Apart from a few very small queries as detailed below, the manuscript, in my opinion, warrants publication.
Introduction
Line 43: please could you add a very short clarification what health-related departments mean? Administrative entities?
Line 47: Please could you revise the sentence. Currently it is a bit difficult to follow.
Line 56: Does the procedure refer to procedure after the exposure and relates to both personal and administrative costs?
Line 68: “Considering these administrative peculiarities…” please consider rephrasing – peculiarities in Beijing or more widely in the system?
Discussion
Line 247: This sentence is difficult to follow as it is not clear immediately to what the percentages at the start of the sentence refer to.
Line 257: “…intervenes…” interventions?